# Study on the Spatial Restructuring of the Village System at the County Level Oriented toward the Rural Revitalization Strategy: A Case of Jintan District, Jiangsu Province

**Xinguo Bu [1,2,3], Lijie Pu [1,3,*], Chunzhu Shen [2,3], Xuefeng Xie [4] and Caiyao Xu [1]**

[1] School of Geography and Ocean Science, Nanjing University, Nanjing 210023, China; dg1427002@smail.nju.edu.cn (X.B.); cyxu@smail.nju.edu.cn (C.X.)
[2] Jiangsu Province Land Surveying and Planning Institute, Nanjing 210017, China; dg1327009@smail.nju.edu.cn
[3] Key Laboratory of Coastal Zone Exploitation and Protection, MNR, Nanjing 210017, China
[4] College of Geography and Environmental Sciences, Zhejiang Normal University, Jinhua 321004, China; xiexuefeng@zjnu.cn
[*] Correspondence: ljpu@nju.edu.cn

**Abstract:** The spatial restructuring of village systems is an important means by which to promote rural revitalization. A large number of villages with small average areas bring great challenges to the implementation of the Rural Revitalization Strategy (RRS) in China. To promote the implementation of the RRS, it is necessary to restructure the village system. This paper proposes a method of spatial restructuring for the village system at the county level, oriented toward the RRS. This study proposes a village classification system with central villages, characteristic villages, and merged villages. It also accounts for the role of various villages in the RRS and proposes differentiated development strategies. This study involved the construction of a village centrality index system and a central village selection model aligned with the RRS. Taking the district of Jintan in Jiangsu Province as a case study for the empirical analysis, the results show that the applicability of the model to the study area is good. Using this model, 32 central villages and 10 characteristic villages were selected. After restructuring the village system, the number of villages decreased by 69.1%. The results from analyzing the travel time radius of the central villages show that 71.5% of the land in the evaluation area lies within a 15 minute commute of the central villages, and 96.5% lies within 25 minutes, indicating that the locations and number of the selected central villages are reasonable. Compared with the service area of the village system before the restructure, the average service area of the central villages is 3.4 times larger, which helps to improve the infrastructure and public service efficiency of the central villages. By guiding resources to aggregate in the central villages and promoting the comprehensive consolidation of land in the merged villages, the restructuring of the village system can help further the success of the RRS in Jintan.

**Keywords:** spatial restructuring; village system; RRS; Jintan District

## 1. Introduction

The problems of agriculture, rural areas, and farmers, known as the "three rural issues", are the outcomes of China's industrialization and urbanization [1–4], which are mainly manifested in the lagging development of agriculture, the decline of the rural economy, the reduction in the rural population, and the widening income gap between urban and rural residents. Over the last few decades, many developed countries and regions have been confronted with similar problems during

their industrialization and urbanization process [5]. To cope with these problems, they have formulated and implemented rural revitalization strategies or relevant policies, such as the "The Common Agricultural Policy" launched by the EU in 1962 [6] and the EU Rural Development Regulation in 2000 [7]; South Korea's Saemaul Undong (New Village Movement), which began in the early 1970s [8,9]; and the Japanese "Village Made Movement" and "One Village One Product" programs in the 1970s [10,11]. Continuous efforts have been devoted to implementing such strategies, which have effectively narrowed the income gap between urban and rural residents and have promoted the coordinated development of urban and rural areas.

The Rural Revitalization Strategy (RRS) of China (referring to mainland China) has been proposed in the process of solving the country's increasingly serious three rural issues. In the 1950s, to quickly establish an independent industrial system and accelerate the industrialization process, China gradually implemented differentiated economic and social policies in urban and rural areas, respectively, including policies on household registration, land, education, medical care, employment, and pensions. Within this context, planned economy methods were adopted to concentrate abundant resources in cities and industries. Thus, an urban–rural dual structure was gradually formed, leading to the gap between the urban and rural development [12,13]. In the late 1970s, China opened its economy, established a socialist market economic system with Chinese characteristics, and strived to break down the urban–rural dual structure. However, the elimination of this dual structure required a long process of integration and shared vision, which could not be achieved in the short term. Hence, the income gap between urban and rural residents continued to widen (Figure 1). By 2009, the income ratio of urban to rural residents in China reached 3.33, and accounting for nonmonetary factors such as public medical care and unemployment insurance, the urban–rural income gap in China ranked the highest in the world [14]. Since 2004, the No. 1 Central Document, issued by the Chinese government annually to outline the economic and social development for that year, has focused on addressing the three rural issues, and has set narrowing the urban–rural gap and promoting the coordinated development of urban and rural areas as the country's top priorities. China has successively implemented a series of supportive policy arrangements, such as the Socialist New Rural Construction [15] and Urban–Rural Coordinated Development [16], which have significantly improved the rural living environment, transportation infrastructure, and living standards of farmers. However, uncoordinated urban and rural development in China remains in a grim situation. In 2018, the national per capita disposable income for urban residents was ¥39,200, while the per capita disposable income of rural residents was ¥14,600 [17]. The ratio of urban to rural residents' income was still as high as 2.69. Moreover, agricultural development was still lagging behind, and the problems of increasing gaps in education, medical care, and health services between urban and rural areas remained to be settled. In November 2017, the 19th National Congress of the Communist Party of China adopted the RRS as one of the seven principal strategies for economic and social development in the coming decades and put forward five overall requirements for China's RRS: booming industries, ecological living environments, a high degree of rural civilization, effective governance, and a wealthy life [18]. To be specific, booming industries are the focus of rural revitalization, through promoting the integrated development of the primary, secondary, and tertiary industries to consolidate the rural economic foundation. Ecological living environments are the primary concern to rural revitalization, which requires the comprehensive management of the rural ecological environment, the enhancement of the supply of agricultural ecological products and services, and the promotion of green rural development. A high level of rural civilization is the guarantee of rural revitalization. With improvements in the rural material civilization, it is also essential to improve the rural spiritual civilization and continuously improve the rural social civilization. Effective governance is the basis for rural revitalization, and a modern rural social governance system should be established to ensure that rural society is characterized by vitality, harmony, and order. A wealthy life is the foundation of rural revitalization, the focus of which is on raising farmers' incomes and improving rural education, medical care, and other public services [19].

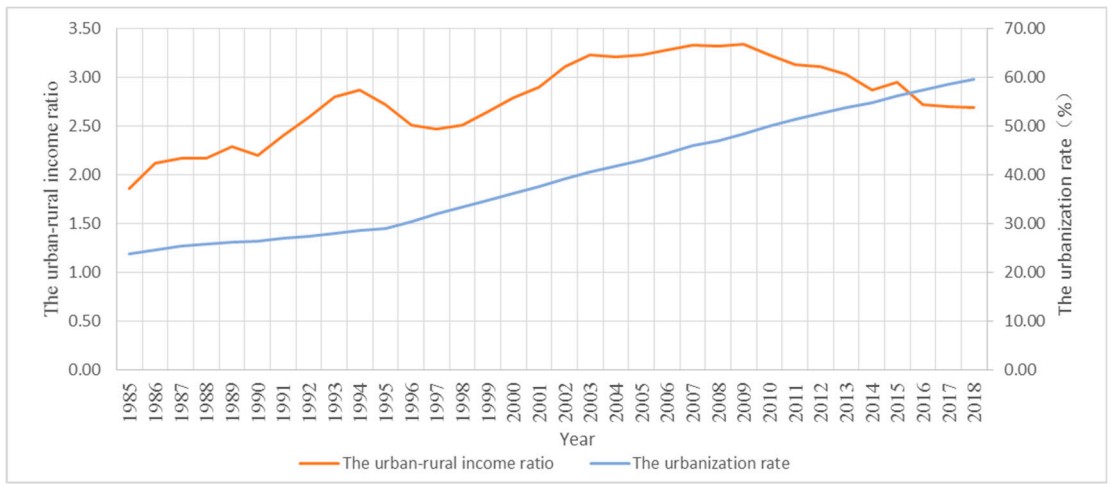

**Figure 1.** The change in the urban–rural income ratio and urbanization rate in China from 1985 to 2018.

Restructuring the village system and guiding the merger of villages are crucial measures for the promotion of rural revitalization. Many developed countries have attached importance to the enhancement of rural revival through the restructuring of village systems. In the second half of the last century, to cope with the hollowing out of the countryside and the tensions between land supply and demand in the process of urbanization and industrialization and to improve the service capabilities of local governments, Sweden, the UK, Germany, and Norway focused on restructuring their village systems. The number of villages decreased by 87.9%, 73.4%, 64.9%, and 39.0%, respectively [20–22]. In other countries such as Canada [23] and the USA [24], village merger programs were also carried out, but the scale of the mergers was relatively small. Furthermore, the village merger in Japan is very representative. After World War II, Japan launched two large-scale mergers of villages to restructure its village system. The first merger, from 1953 to 1961, was called the "great Showa merger", and the second from 1999 to 2006 was named the "great Heisei merger". Through these two mergers, the number of villages in Japan decreased from 10,411 in 1950 to 1730 in 2010 [25], a reduction of 83.4%. By restructuring its village system, Japan constructed central village infrastructure and public service facilities to agglomerate rural housing and populations and then launched the "Village Made Movement" and the "One Village One Product Movement". Rural revitalization in Japan significantly improved the quality of life of rural residents; in 1974, the income of rural residents exceeded that of urban residents for the first time. The income ratio of urban to rural residents in Japan currently fluctuates between 0.86 and 0.97 [26].

In 2018, China had 2851 county-level administrative units [17], with approximately 680,000 village-level administrative units and 239 village-level administrative units per county. The average area of the village-level administrative units was approximately 14 km², equivalent to 6.5% of that of Japanese villages (approximately 218 km²). A large number of village-level administrative units bring great challenges to the implementation of the RRS in China. On the one hand, scattered villages lead to redundant investments in the construction of rural infrastructure and public service facilities, which will impose a heavy burden on finances. On the other hand, complicated local organizations reduce the efficiency of village governance. Thirdly, a tremendous number of administrative villages results in rural construction land decentralization, which further contributes to the fragmentation of agricultural land and ecological land. Under this circumstance, great obstacles have obstructed the large-scale agricultural operation and the improvement of the rural ecological environment. Hence, to promote the implementation of China's RRS, it is imperative to restructure the village system and reduce the number of villages significantly. This study attempted to explore the restructuring of the village system at the county level based on a village centrality evaluation model aligned with the RRS, and this paper provides academic support for the implementation of the RRS.

## 2. Methods for the Spatial Restructuring of the Village System

## *2.1. Village System Composition and Development Strategies*

The village is an important link in the development of settlements and an important center of rural life and production. The village system refers to the organic whole composed of villages of different levels and functions within a certain geographical region. The spatial restructuring of the village system refers to the selection of different types of villages according to the current conditions of the villages and the rearrangement of the village structure's hierarchical system. There are several categories of village system composition. The Town Planning Standards (GB50188-2007) issued by China in 2007 divided villages into central villages and local villages based on population. The population of central villages is between 300 and 10,000 people, and the population of local villages is less than 300 [27]. Based on the direction of the future development of the villages, Sun (2009) divided the villages into three types: urbanization villages, relocation villages, and developing villages [28]. Based on village functions, Chen (2017) divided villages into four types: key villages, characteristic villages, general villages, and small/scattered villages [29]. Based on the spatial restructuring of the village system, this study divides villages into three types: central villages, characteristic villages, and merged villages.

Central villages are villages with good central characteristics that can provide quality services to other surrounding villages and have advantages in terms of transportation, location, industrial foundation, public services, and natural environment. Central villages are the main vehicle for the realization of the overall requirements of the RRS. Central villages are important areas for rural population agglomeration; for the integrated development of the primary, secondary, and tertiary industries; and for the construction of rural housing, rural infrastructure and public services and public management facilities in the implementation of the RRS. The development strategy for central villages is to rationally plan the use of village land space; strengthen the construction of infrastructure and public service facilities, such as facilities for education, medical care, culture, and elderly care; develop secondary and tertiary industries suitable for rural areas; and attract villagers from merged villages to settle and find employment.

Characteristic villages refer to villages with characteristic landscapes and historical and cultural heritages that have value and should be protected. They are mainly composed of well-preserved and well-developed ancient villages and characteristic tourist villages. Characteristic villages are important areas for the development of rural tourism as part of the RRS. The development strategy for characteristic villages is to adhere to the principle of placing equal emphasis on protection and development, fully utilizing the advantages of characteristic villages in terms of natural ecology, history and culture, characteristic industries, and folk customs to make characteristic villages a recreational tourist attraction for urban residents.

Merged villages are villages other than the above two types. There are two subtypes of merged villages. The first subtype is the suburban merged village, which is mostly or entirely within the urban planning area; the second subtype is the hinterland merged village, which is far away from cities and towns, with relatively poor development conditions and a low centrality index. Suburban merged villages are important components of urban development, and their development strategy is to gradually upgrade and transform in accordance with urban planning and to integrate their development with that of the central city. The hinterland merged villages are important areas for the development of large-scale, characteristic, and specialized agriculture, for which the development strategy is to gradually reclaim idle and inefficient rural construction land and consolidate agricultural land so as to promote the concentration of farmland and improve the ecological network under the guidance of spatial planning.

## *2.2. Village Centrality Index System and Evaluation Model*

With the goal of assisting in the realization of rural revitalization and guided by the overall requirements of China's RRS, a village centrality evaluation index system was constructed to quantitatively evaluate village centrality characteristics. Central villages are the main locations of rural population residence, which should have relatively good environmental conditions and good transportation locations, public services, and public management conditions to improve the quality

of life of residents. Central villages are also the center of rural industrial revitalization, so these villages should also have a good industrial foundation to provide residents with diversified employment opportunities. In general, at the county level, the factors affecting village centrality can be divided into two categories: environmental conditions and socioeconomic conditions. The environmental conditions mainly include the terrain, ecological livability, water source guarantees, and natural disasters. Socioeconomic conditions mainly include the population size, transportation accessibility, degree of public service, public management support, and the industrial development foundation. The relationship between the evaluation factors, the indicators, and the RRS are shown in Table 1.

**Table 1.** The evaluation index system for village centrality.

| Condition | Determinants | Optional evaluation indicators | The relationship between the determinants and the Rural Revitalization Strategy (RRS) |
|---|---|---|---|
| Environmental conditions | Terrain | Average elevation, average slope | Reflects the basic conditions for the development of villages. For areas with large hills or mountains, the topographical condition is an important determinant of the choice of central villages, while for the plains, this factor can be ignored. |
| | Ecological livability | Proportion of ecological land | Reflects the ecological livability of villages, and indicates the proportion of water areas, forest land and other ecological land in the area. |
| | Water source guarantee | Proportion of water area | Water sources are the lifeblood of rural areas and agricultural production. For water resource-constrained areas, the water source guarantee rate is an important determinant of central villages. For areas along a water network, this factor can be ignored. |
| | Natural disasters | Areas severely affected by natural disasters. | Various areas prone to natural disasters are automatically disqualified in the site selection of the central villages. |
| Socioeconomic conditions | Demographic status | Population density | Reflects the village's ability to attract residents. Villages with severe hollowing out and a sparse population are not suitable to be central villages. |
| | Transportation accessibility | Commuting time to nearest town | Reflects the ease of transportation of the villages. Central villages are the links between the merged villages, the characteristic villages, and the towns, which requires that these villages are relatively easily accessible via transportation. |
| | Public management and services support | Proportion of public services and public management land | Convenient public services and management can attract residents to settle and help improve the quality of life of residents. |
| | Nonagricultural industry base | Proportion of collective commercial construction land | Reflects the basis for integrated development of the village industries. The larger the nonagricultural industry base is, the better the ability of the village to absorb population and |

| | | to promote industrial agglomeration, and the more prominent the centrality of the village. |
|---|---|---|
| The base for large-scale agricultural operations | Agricultural land contract management rights transfer ratio | Reflects the basic conditions for the development of modern agriculture in the village. A high transfer ratio of agricultural land contracted management rights provides a good foundation for large-scale operations, and these types of villages are more suitable to be merged villages. |
| Degree of idle construction land | The proportion of idle rural construction land | Reflects the utilization rate of rural construction land. Villages with a high rate of idle rural construction land are not suitable to be central villages. |

The village centrality model is as follows:

$$C_i = \sum_{i=1}^{n} \sum_{j=1}^{m} M_i * W_j \qquad (1)$$

In the formula, $C_i$ represents the centrality of village *i*; $M_i$ is the evaluation index value; $W_j$ is the index weight, which is determined by the structural entropy weight method; *n* is the number of evaluated villages; and *m* is the number of evaluation indicators, which include 4–6 items selected from Table 1 according to the local natural, social, and economic conditions and the availability of the indicators.

The centrality index for each village is calculated by the model. The villages with a high index value can be regarded as central villages. Moreover, villages with landscapes of important historical and cultural value or scenic tourist landscapes should be retained as characteristic villages, even if their centrality index is relatively low. With reference to Christaller's central place theory [30,31] and determining the number of central places based on market and transportation principles, the hierarchical system of central places (from the highest level to the lowest level) is roughly as follows: grade A: 1, grade B: 2–3, grade C: 6–12, and grade D: 18–48. For a county, the central place system generally constitutes a hierarchical structure of central city (grade A), key town (grade B), general town (grade C), and central village (grade D). Therefore, the number of central villages was set at 18–48.

## 3. Study Area and Data Source

### 3.1. Overview of the Study Area

Jintan District is located in Eastern China, in the hinterlands of the Yangtze River Delta in Southwestern Jiangsu Province, and is under the jurisdiction of Changzhou. Its borders are Nanjing to the west, Zhenjiang to the north, Liyang to the south, Wujin to the northeast, and Wuxi to the southeast (Figure 2). Jintan is a typical socioeconomically developed and densely populated area in Eastern China. In 2018, Jintan District had a total land area of 975.46 km², a population of 562,000, an urbanization rate of 62.5%, and a population density of 576 people/km²; the annual GDP was ¥80.193 billion, and the per capita GDP was ¥142,819 (approximately $21,600). The annual per capita disposable income of urban residents was ¥50,770, that of rural residents was ¥26,283, and the income ratio of urban to rural residents was 1.91:1. The income gap between urban and rural residents in Jintan was still large.

Jintan is located in the subtropical monsoon climate zone with four distinct seasons, an average annual temperature of 15.3°C, and an annual precipitation of 1063.5 mm. The terrain in Jintan slopes from west to east, with Mao Hill in the west accounting for approximately 20% of the land area, and

the highest elevation is 372.5 m; the east is a low-lying plain, accounting for approximately 80% of the land area. There is a dense water network in the territory, with a lake named Changdang Lake in the south, which covers an area of 82 km². Jintan has various territorial features, such as hills, water, forests, fields, lakes, and towns. On the whole, the natural and socioeconomic characteristics of Jintan are reasonably representative and are suitable as a case study area.

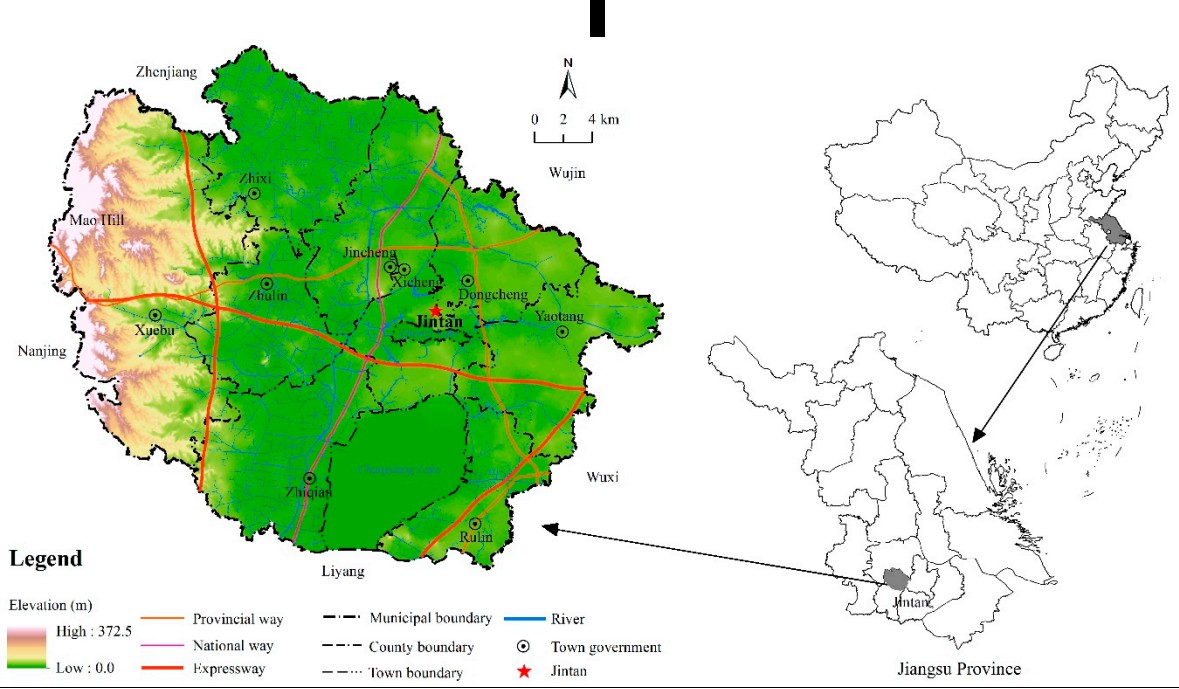

**Figure 2.** Location and basic geographic information on the study area, Jintan District.

In 2018, Jintan District governed nine towns: Yaotang, Dongcheng, Xicheng, Jincheng, Rulin, Zhixi, Zhulin, Xuebu, and Zhiqian. There are 171 village-level administrative units within the district. Excluding the units involved in lakes, the contiguous developed area of the central city, and the villages where the town governments are located, there are 136 administrative villages in Jintan, with an area of 751.17 km², and the average village area is 5.57 km².

### 3.2. The Centrality Index and Data Source

According to the characteristics of the natural environment of Jintan District, the chosen indicators for natural conditions were the average slope (a) and the proportion of ecological land (b). According to the availability of indicators, the socioeconomic factors chosen were population density (c), agricultural land transfer ratio (d), commuting time to the nearest town (e), and public service and management land ratio (f). The average slope was calculated using 1:50,000 DEM (Digital Elevation Model) data in Jintan District. The proportion of ecological land used is the ratio of the area of rivers, beaches, and woodland to the total land area. The commuting time to the nearest town was calculated with the cost distance model in ArcGIS. The cost coefficient was calculated using the speed of the different transportation types available in the area. Based on road speed limits and actual driving experience, the speed of the transportation network is shown in Table 2. The transportation speed of the water areas was set to 2 km/h, and that of other area was 5 km/h. The land use data are from the land use survey of Jintan in 2018, and the socioeconomic data come from the Statistical Yearbook of Jintan District [32].

**Table 2.** Speed of the transportation network in Jintan District.

| Transportation types | Expressway | National way | Provincial way | County road | Urban road |
|---|---|---|---|---|---|
| Speed（km/h） | 120 | 80 | 60 | 40 | 20 |

## 4. Results and Discussion

### 4.1. The Centrality of Villages in Jintan District

The indicators for the 136 villages and their classifications are shown in Figure 3. In general, the influence of natural conditions on the centrality of the village is strongest in the western hilly area, where the average slope is relatively large. There are five villages with an average slope of more than 5°, all of which are located in Xuebu Town (Figure 3a), and the largest slope is 13.5°. The ecological land in Jintan is mainly composed of woodland and river wetlands, and these are most concentrated in the western hilly area (Figure 3b), where woodland accounts for more than 40% of all land in some villages. The population density of villages is generally higher in the east and lower in the west (Figure 3c) Villages with a population density higher than 600 persons/km$^2$ are mainly located in Yaotang and Zhixi. The population density of villages in the Mao Hill area is generally low. Jintan district has relatively high levels of transportation accessibility; most villages' commuting times to the nearest town are less than 20 minutes, and villages with commuting times of more than 25 minutes are mainly located in the northwest and north (Figure 3e). The distribution of the agricultural land transfer ratio (Figure 3d) and public service and management land ratio (Figure 3f) in the whole region is not very significant. Range standardization was used to normalize the six indicators of the 136 villages participating in the evaluation, and the weights of the indicators were determined by the structural entropy weight method, resulting in weights of 22.8%, 11.0%, 7.2%, 12.5%, 5.5%, and 41.0%, respectively.

Formula 1 was used to calculate the centrality index for each administrative village. The average centrality index of the 136 villages is 27.8. The village with the lowest centrality index is the village of Xinhua in Rulin, and its index is 10.5. The village with the highest centrality index is the village of Shuibei in Yaotang, and its index is 63.6. There are 81 villages for which the centrality index is less than the average value, and these are mostly concentrated in the western region of Jintan. Fifteen villages with an index of 27.8–30 are mainly distributed in the central region. Twelve villages with an index of 30–35 are mainly distributed in Zhulin and Zhiqian. Twenty-eight villages with an index greater than 35 are widely distributed around towns and in rural hinterlands (Figure 4).

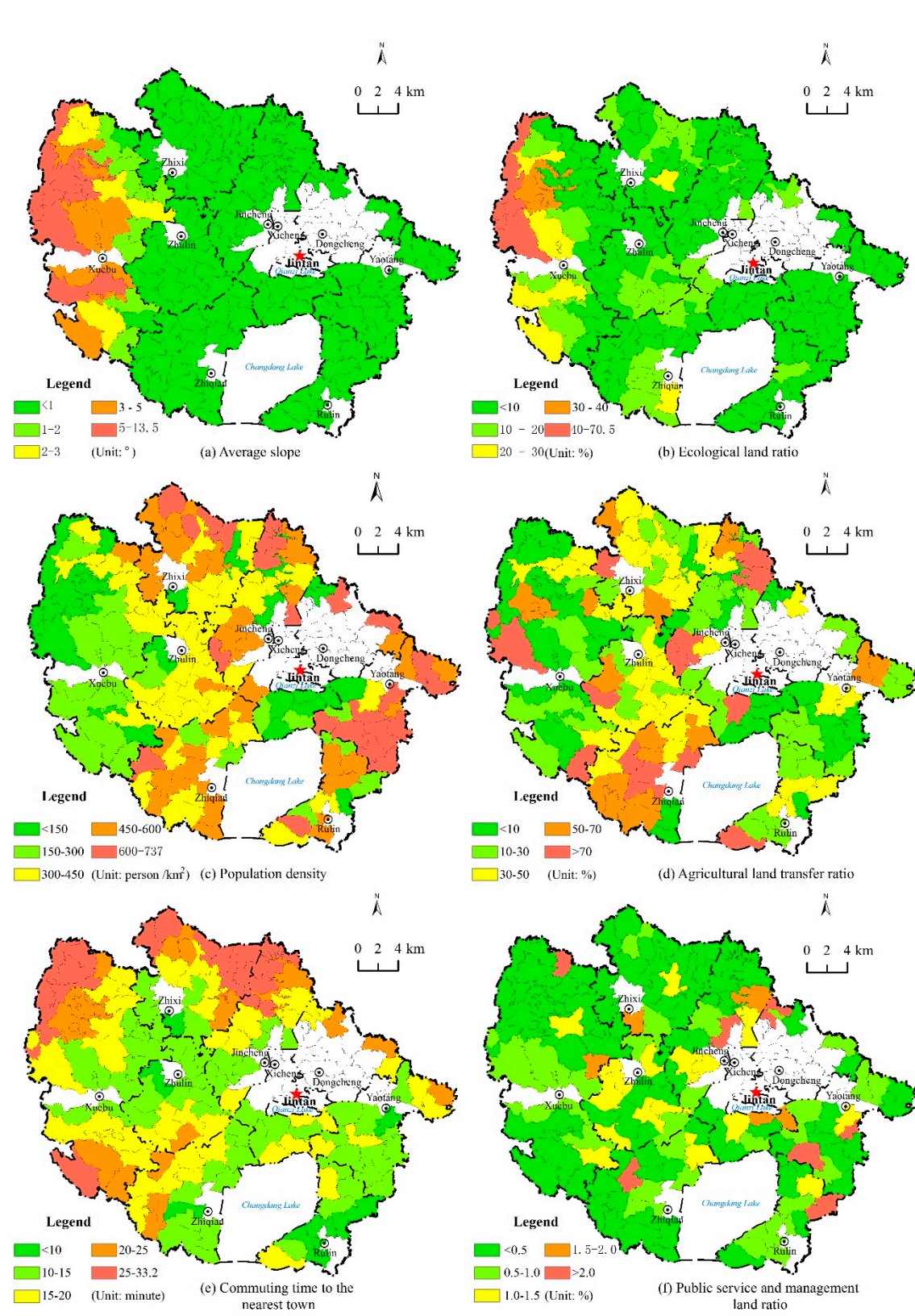

**Figure 3.** Centrality indicators for villages in Jintan District.

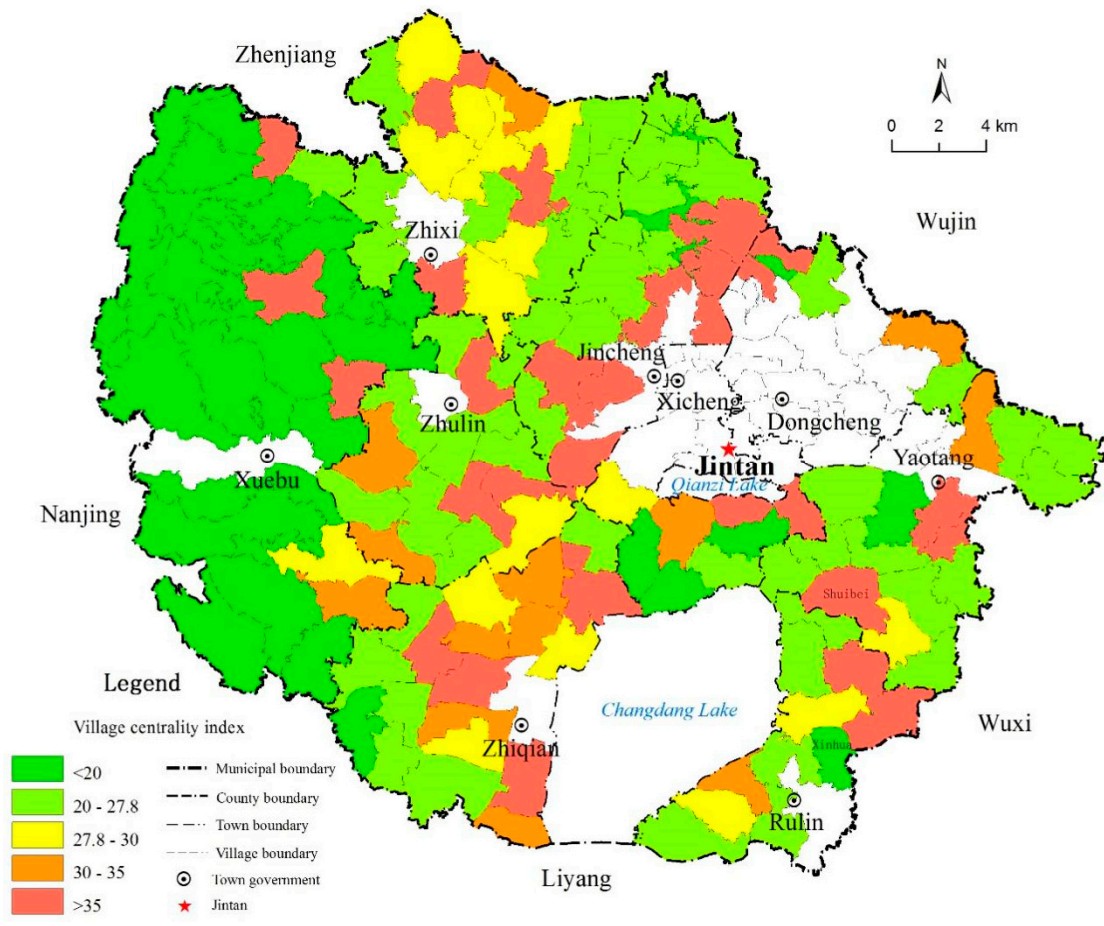

**Figure 4.** The centrality index of each village in Jintan District.

*4.2. The Restructured Village System of Jintan District*

The central villages were determined according to the following three rules. First, the centrality index is the main basis for identifying the central village. Commonly, villages with a centrality index higher than the average value are eligible to become central villages. When multiple neighboring villages meet this condition, the village with the higher index value is preferred. The second is that if most of the land of a village is located within a central urban planning boundary, regardless of whether its index is high or low, it is regarded as a merged village. Third, villages where most of the land is located in areas prone to natural disasters, regardless of their centrality index, are not classified as central villages.

Of the 136 villages participating in the evaluation in Jintan, 45 have a centrality index higher than the average. According to the above rules, 32 central villages in Jintan were selected, and their centrality indexes are all above 30. Of the 13 villages not selected, four are located within the central city planning boundary area [33], and the remaining nine villages were removed based on the first rule. The type of natural disaster to which villages are prone in Jintan is karst collapse, which mainly occurs in the hilly area of the northwest and does not occur in the selected central villages. Jintan has a long history and beautiful natural environment. It has rich pastoral landscapes and famous historical cultural villages, including Xiangu, Duida, Zhihe, Shangyang, and Shangruan in the town of Xuebu; Wangmuguan and Dongpu in the town of Zhiqian; and Yushan, Hutou, and Houzhuang in the town of Rulin. These villages are key villages for rural tourism in Jintan, and they all serve as characteristic villages.

Based on the above analysis, the village system in Jintan was identified and is composed of 32 central villages, 10 characteristic villages, and 94 merged villages, as shown in Figure 5.

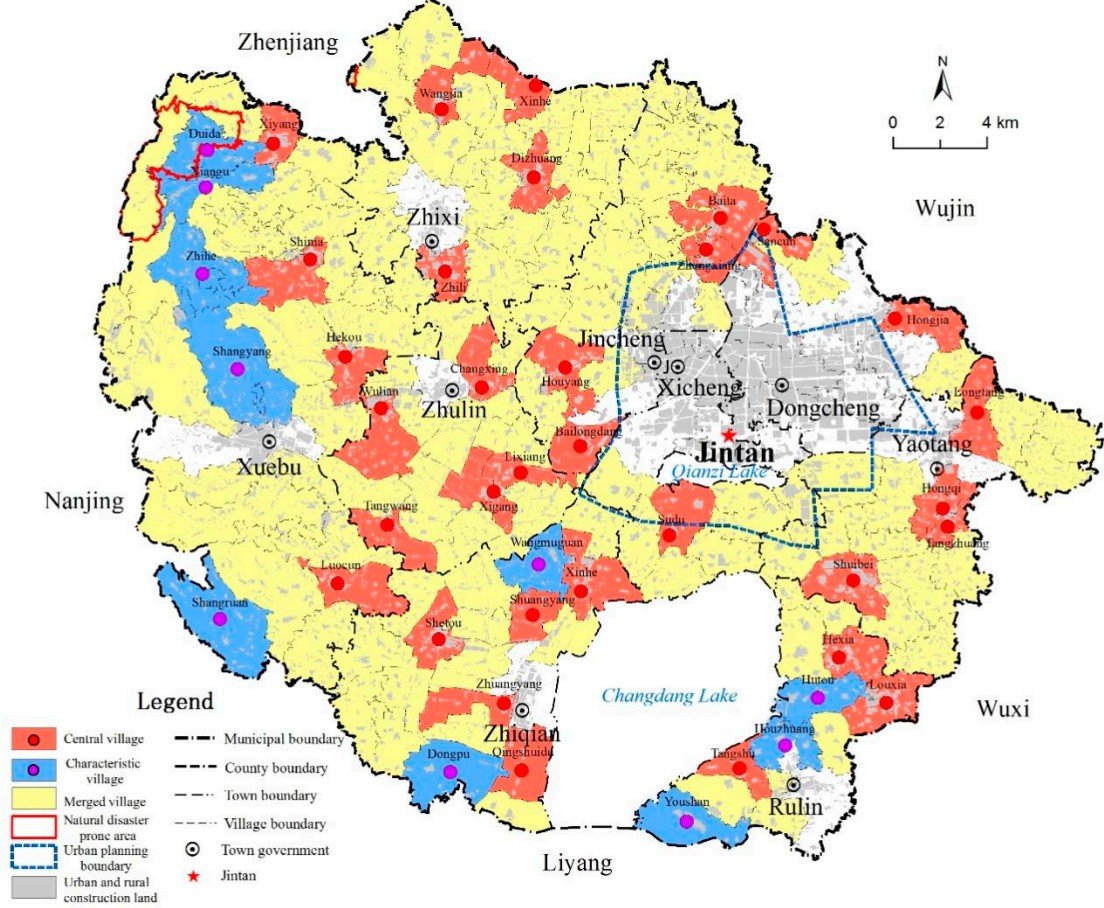

**Figure 5.** The village system in Jintan District.

*4.3. Analysis of the Layout Rationality of the Central Villages in Jintan District*

The cost distance model was used to analyze the travel time radius of the central villages using the locations of the village government stations as the source points, which form a map of the distribution of travel time radii of the center villages in Jintan. According to Figure 6, due to the excellent transportation infrastructure and high accessibility of Jintan, the land in the evaluation area that lies within a 15 minute commute of the 32 central villages is 538.7 km², accounting for 71.5%; that within 20 minutes is 672.7 km², accounting for 89.3%; and that within 25 minutes is 727.1 km², accounting for 96.5%. Areas that take more than 25 minutes to reach are mainly located on Mao Hill. In general, the location and number of central villages are reasonable, which ensures the appropriate farming radius for large-scale agricultural operations.

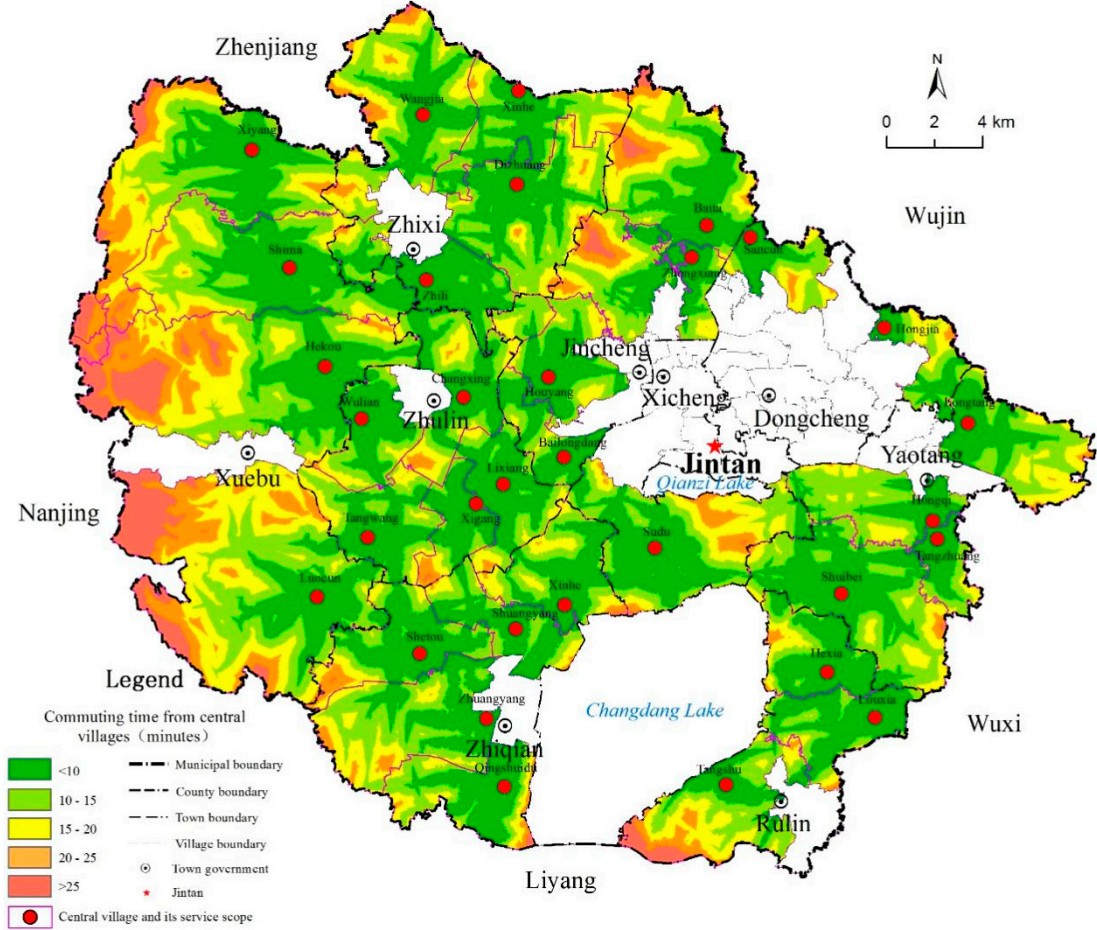

**Figure 6.** Central villages' commuting time and service scope in Jintan District.

The villages falling within the 15 minute travel range of the central village are taken as lying within its service scope, but this scope does not cross town boundaries. If the same merged village falls within more than two central villages' service scopes, its attribution is determined by the principle of proximity. Through the village merger, the service scope of the 32 central villages (including the characteristic villages nearby) is formed. After restructuring the village system, the average service area of the central village is 23.55 km², of which the largest central village is Luo in the town of Xuebu with an area of 61.21 km², and the smallest is the village of Tangzhuang in Yaotang with an area of 6.90 km². Compared with the average service area of the villages before the spatial restructuring, the service area of the central village is 3.4 times larger. Through the spatial restructuring of the village system, the number of villages in Jintan has been reduced by 69.1%, which would help to further improve the level of rural governance.

### 4.4. Suggestions for the Implementation of the RRS in Jintan District

The 32 central villages of Jintan have advantages in terms of their natural environment, economic foundation, location conditions, etc. The governments should concentrate the resources needed for public management and public services, infrastructure construction, and industrial development in the central villages, which would improve the efficiency of resource use. Strengthening the construction of central villages and guiding the aggregation of the rural population and secondary and tertiary industries suitable for rural areas to central villages will help to improve the rural living environment; promote the integrated development of rural primary, secondary, and tertiary industries; expand employment channels for the rural population; increase income levels; and narrow the income gap between urban and rural areas.

The 10 characteristic villages are essentially historical and cultural heritage protection regions and rural tourism destinations in Jintan District. The Mao Hill is a famous Taoist hill in China, with rich Taoist historical buildings and intangible culture. The four characteristic villages located in the Mao Hill are required to balance the relationship between the protection and development of distinctive culture to promote development with protection, making the traditional historical culture a crucial source of material wealth and spiritual wealth for local residents. The other characteristic villages should fully take advantage of folk customs and natural scenery, conduct environmental village improvements and engage in tourism infrastructure construction, and provide unique leisure and entertainment venues for the urban residents. Furthermore, these coordinative development modes provide more opportunities for local residents to enhance their income level and living standards.

The merged villages are pivotal areas for the development of modern agriculture. Under the premise of respecting the wishes of rural residents, the local governments should guide the rural residents in the merged villages to gather in cities and central villages nearby. In the four suburban merged villages, the parts located within the urban planning boundary should develop with the central city, while the other parts can seek to develop characteristic agriculture, such as experience agriculture and sightseeing agriculture. The 90 hinterland merged villages should carry out comprehensive land consolidation, gradually reclaim the idle and inefficient rural construction land to improve the concentration of farmland and connections among ecological corridors, and promote the development of modern agriculture and improve the rural ecological environment.

In general, by guiding differentiated developments in the central villages, characteristic villages, and merged villages, the spatial restructuring of the village system can promote the implementation of the RRS in Jintan District.

## 5. Conclusions and Future Work

The village is an important center for rural production and life, and central villages are the focal and key points for the implementation of the RRS. In the process of formulating and implementing their rural revitalization strategies and similar policies, many developed countries have attached great importance to improving the level of rural governance by restructuring their village systems, stimulating rural development, and promoting the coordinated development of urban and rural areas. China's RRS has been proposed as part of the process of solving the increasingly serious three rural issues. The number of villages in China is a major problem for the implementation of the RRS. It is imperative to restructure the village system and reduce the number of villages significantly.

This paper proposes a method for spatially restructuring the county-level village system in line with the RRS and validates it with a case study of Jintan, Jiangsu Province. This study proposed that the villages should be divided into central villages, characteristic villages, and merged villages and that each village type should implement differentiated development strategies. Each type of village plays a different role in the implementation of the RRS. In accordance with the overall requirements of the RRS, this paper proposes an index of village centrality based on natural conditions and socioeconomic conditions, and a model was constructed to select central villages. The case study results show that the applicability of the model to the study area is good. Using this model, 32 central villages and 10 characteristic villages were selected in Jintan. After restructuring the village system, the number of villages was reduced by 69.1%. A 15 minute travel radius from the central villages in Jintan covers 71.5% of the evaluation area, and a 25 minute travel radius covers 96.5% of the land. Therefore, the locations and number of the central villages are reasonable. Compared with the service area of the village system before restructuring, the average service area of the central villages is 3.4 times larger, which helps to improve the infrastructure and public service efficiency of the central villages. Generally, by guiding resources to aggregate in the central villages and promoting the comprehensive consolidation of land in the merged villages, the restructured village system can help the RRS succeed in Jintan.

Three aspects of this research need to be further explored. First, the index selection of the central evaluation model needs to be further improved. China's rural areas are vast, and the natural

environment and socioeconomic conditions are very different across regions. The problems faced by counties in different regions in implementing the RRS are complex and changeable. Some of the evaluation indexes need to be changed to align with actual conditions. Therefore, the selection of evaluation indicators in different regions needs further research. Second, the applicability of the model needs further verification. The study area selected for this analysis has both hills and plains, with a developed economy and a dense population. The final selection of central villages reflects the influence of natural and socioeconomic conditions on the centrality of the villages, and the applicability of the model is good. However, for counties located in hilly, mountainous, and plateau regions, as well as counties with underdeveloped economies and relatively sparse populations, the applicability of the model needs to be further verified and revised. Third, the influence of certain specific factors on the selection of central villages needs further study. For example, in counties where many ethnic groups live, the lifestyles and religious beliefs of different ethnic groups will affect the merger of villages. Additionally, the influence of the villagers' land preferences may affect the choice of central villages.

**Author Contributions:** All authors contributed to data collection. Conceptualization, X.B.; project administration, L.P.; writing—original draft, X.B.; writing—review and editing, L.P., C.S., X.X. and C.X. All authors have read and agreed to the published version of the manuscript.

**Funding:** This research was funded by National Science and Technology Support Program (No. 2015BAD06B02) and National Natural Science Foundation of China (No.41871083).

**Conflicts of Interest:** The authors declare no conflict of interest in this paper and study.

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
