# Peer review of "Study on the Spatial Restructuring of the Village System at the County Level Oriented toward the Rural Revitalization Strategy: A Case of Jintan District, Jiangsu Province"

_land, doi:10.3390/land9120478_

Round 1

Reviewer 1 Report

The article has been improved and I congratulate the authors for the effort. Now it is better explained and their proposal has scientific and planning interest. 

Authors should be notice that many references give an error (Error! Reference source not found). 

The introduction must be improved. First, there is a misunderstanding with verb tenses and the use of present, past, past perfect, etc. (just in the first section, methods, results and conclusions are fine). 

In the first paragraph it is said "Most developed countries have faced similar problems in their development process". Please specify which countries and/or give references. Then, authors talk about developing countries but the first two examples are from EU. 

I don't understand this sentence "the South Korean New Village Movement begun in the early 1970s, rural revitalization in the 1960s, and..." I think they are out of context or should be re-written. 

In line 89-90, concepts such as "rural material civilization", "rural spiritual civilization" and "rural social civilization" need an explanation. 

In line 92 "a modern rural social governance system has been established to ensure that rural society is full of vitality, harmony and order. How can be vitality, harmony and order measured? Vitality and harmony depends on the people's perceptions, these concepts are vague. 

Line 125:  "scattered villages require repeated investments in the construction of rural infrastructure and public service facilities, which is a serious waste" Needs a citation or the use of another kind of word but not "serious waste"

Figures 2 and 3 should be improved. I cannot see the legends. 

Finally, there is not a discussion, but only a conclusion. 

Author Response

November 26, 2020

Dear reviewer:

Thank you very much for your professional review work on our manuscript. We have addressed the comments and the amendments are highlighted using the "Track Changes" in the revised manuscript. Point by point responses to the comments are listed below this letter. This manuscript has been edited by Academic Journal Experts for English language.

We hope that the revised version of the manuscript is now acceptable for publication in your journal.

I look forward to hearing from you soon.

With best wishes,

Yours sincerely,

Xinguo Bu

The replies are as follows:

Comment 1:Authors should be notice that many references give an error (Error! Reference source not found). 

Answer:Sorry for having many reference problems. We have checked that some of the references cannot be found because they are Chinese papers. We have tried our best to replace these references with English references that are similar to the original ones. The numbers of these new references are 3, 8, 9, 10, 11, 25, and 26. We have kept some Chinese references which we could not find replaceable ones, such as some policy files, books, Master and PhD thesis, and so on.

Comment 2:The introduction must be improved. First, there is a misunderstanding with verb tenses and the use of present, past, past perfect, etc. (just in the first section, methods, results and conclusions are fine).

Answer:We have improved greatly the English language in the part of introduction, including verb tenses and other expressions.

Comment 3:In the first paragraph it is said "Most developed countries have faced similar problems in their development process". Please specify which countries and/or give references. Then, authors talk about developing countries but the first two examples are from EU. 

Answer:We have revised this sentence to “Over the last decades, many developed countries and regions have ever been confronted with similar problems during their industrialization and urbanization process” (line 39-41), and have provided a reference, titled “The Comparison and Inspiration of International Experience in Rural Revitalization: Taking Japan, South Korea and EU as Examples” (reference 5).

Comment 4:I don't understand this sentence "the South Korean New Village Movement begun in the early 1970s, rural revitalization in the 1960s, and..." I think they are out of context or should be re-written. 

Answer:We have revised this sentence as “the South Korean New Village Movement begun in the early 1970s, and the Japanese “Village Made Movement” and “One Village One Product” programs in the 1970s”.

We had written the “rural revitalization in the 1960s”, because we hoped to express that Japan began to implement its rural revitalization strategy in the 1960s. In 1961, Japan lunched the “Japanese Agriculture Basic Act”, which was an important law for the rural revitalization in Japan. But Japanese “Village Made Movement” and “One Village One Product” programs are more widely known. So we deleted the “rural revitalization in the 1960s”.

Comment 5:In line 89-90, concepts such as "rural material civilization", "rural spiritual civilization" and "rural social civilization" need an explanation. 

Answer:China generally divides social civilization into material civilization and spiritual civilization. Material civilization refers to the progress of human material life, which is mainly manifested in the progress of material production methods and economic life. Spiritual civilization refers to the spiritual wealth created during the historical practice of human society, including ideology, morality and education, science, culture, etc.

Comment 6:In line 92 "a modern rural social governance system has been established to ensure that rural society is full of vitality, harmony and order. How can be vitality, harmony and order measured? Vitality and harmony depends on the people's perceptions, these concepts are vague.

Answer:The Chinese No.1 Central Document in 2018 proposed that “adhere to the combination of autonomy, rule of law, and rule of virtue to ensure that rural society is full of vitality, harmony and order”. The concepts of vitality, harmony and order in this manuscript are from this document. Chinese government believes the measurement signs of vitality, harmony and order are freedom, equality, fairness, and the rule of law. However, these concepts are relatively abstract, and the Chinese government has not yet formulated quantitative indicators. We think that your question about how to measure them is forward-looking.

Comment 7:Line 125:  "scattered villages require repeated investments in the construction of rural infrastructure and public service facilities, which is a serious waste" Needs a citation or the use of another kind of word but not "serious waste".

Answer:We have rewritten this sentence to “scattered villages lead to redundant investments in the construction of rural infrastructure and public service facilities, which will impose a heavy burden on the finances”.

Comment 8: Figures 2 and 3 should be improved. I cannot see the legends

Answer:It may be that the figures were compressed in the word document, which made the legends unclear. We have provided these figures’ files with this manuscript.

Comment 9: Finally, there is not a discussion, but only a conclusion.

Answer:We have written the section of discussion in Part 4 “Results and Discussion”, in which we discussed the suggestions for the implementing of the RRS in Jintan District. The title of Part 5 has been revised to “Conclusions and Future Work”.

Reviewer 2 Report

The paper “A Study on the Spatial Restructuring of the Village System at the County Level and Oriented toward the Rural Revitalization Strategy: A Case Study of Jintan District, Jiangsu Province” has been much improved. Few suggestions prior to be published:

  • A more concise title
  • References throughout the paper should be fixed
  • The aim of the paper should be explicitly described
  • A clearer description of the methodology followed, and the methods applied should be inserted at the very beginning of the methodology section.
  • Discussion should preceed the conclusion

Author Response

November 26, 2020

Dear reviewer:

Thank you very much for your professional review work on our manuscript. We have addressed the comments and the amendments are highlighted using the "Track Changes" in the revised manuscript. Point by point responses to the comments are listed below this letter. This manuscript has been edited by Academic Journal Experts for English language.

We hope that the revised version of the manuscript is now acceptable for publication in your journal.

I look forward to hearing from you soon.

With best wishes,

Yours sincerely,

Xinguo Bu

The replies are as follows:

Comment 1:A more concise title

Answer:We have revised the title to “Study on the Spatial Restructuring of the Village System at County Level Oriented toward the Rural Revitalization Strategy: A Case of Jintan District, Jiangsu Province”

Comment 2:References throughout the paper should be fixed

Answer:We have improved the references by inserting “cross reference”, and all references are fixed.

Comment 3:The aim of the paper should be explicitly described

Answer:We have described the aim of the paper: “This paper attempts to explore the restructuring of the village system at the county level based on a village centrality evaluation model aligned with the RRS and provides academic supports for the implementing of the RRS.”

Comment 4: A clearer description of the methodology followed, and the methods applied should be inserted at the very beginning of the methodology section.

Answer:This paper constructs a village centrality index system and a central village selection model aligned with the RRS,we have elaborated on the method of spatial restructuring of the village system in the part 2.

Comment 5: Discussion should preceed the conclusion

Answer:We have written the section of discussion in Part 4 “Results and Discussion”, in which we discussed the suggestions for the implementing of the RRS in Jintan District. The title of Part 5 has been revised to “Conclusions and Future Work”.

This manuscript is a resubmission of an earlier submission. The following is a list of the peer review reports and author responses from that submission.

Round 1

Reviewer 1 Report

The paper “Study on the Spatial Restructuring of Urban-Rural Construction Land Oriented towards the Rural  Revitalization Strategy” aims to support the RRS implementation in China. Specifically, the authors propose a method to guide the agglomeration and the orderly and efficient use of urban-rural construction land including two core links: the restructuring of the city-town-village system and the delimitation of the urban-rural growth boundaries.

There are several shortcomings that should be addressed in order the paper to be published:

First of all the authors are kindly invited to clearly state their aims at the beginning of the paper. Lines 160-161 and then lines 177-180 depict the objectives of the research but they need to be inserted at the very beginning.

The discussion about European rural restructuring (references 7-10) are somehow misleading: they refer to a sociological perspective rather than a spatial and/or land perspective. This should be clarified both in terms of concepts and in terms of contribution to the Chinese experience.

In general the impression I got reading the paper is that the authors go all around but they are not on the point. Till the paragraph 3 the reader is accompanied in a journey through theories and approaches without any guide….Maybe this discussion could follow the introduction of the methodology framework that is provided in paragraph 3.

Minor suggestion:

  • Line 36 “the rural revival of the EU in the 1960s”: what does it refer to?
  • Check lines 170-172

Reviewer 2 Report

The authors present an interesting research to improve the Rural Revitalization Strategy in China regarding urban-rural construction land. It is a theoretical research of a topic of great interest. However, it needs to be improved and to deepen into the different concepts and subjects they present.

In the abstract it is said: “This study examines the spatial restructuring of urban-rural construction land, provides theoretical and methodological support for solving the land use problems faced by the RRS.” The spatial restructuring of urban-rural construction is not really examined. They talk about some problems but do not deepen into it. The methodological approach has not been applied so we don’t know if it works or not.

They make some suggestions without really explaining why they choose them (and not others) and above all, if these propositions will really work to solve the problem, like: remove of barriers through land reclamation to improve the agriculture scale operation (but on the other hand, leisure-sightseeing agriculture or ecological tourism – the authors do not specify, for example, how to combine these) or more land available for construction in China arguing that is already very limited and other developed countries have a highest urbanization rate. Also, in lines 340 it is said that “By restructuring the urban-rural construction land space and guiding the rural population to gather in cities, towns and central villages, spatial restructuring can concentrate agricultural land in the hands of large production households, help control the use of pesticides and fertilizers, eliminate rural pollution, promote ecological green agriculture development, and improve rural ecological networks. Thereby, the ecological environment can be greatly improved” The reviewer ask himself: how is going to be implement all this and all this would help to solve the problem?, for example: concentrating agriculture land in the hands of large production households is related better promotion of ecological green agriculture and a control of pesticides? How? What evidences of this has the authors?

In part 4, authors present a methodological proposal, with some index, models and calculations to improve the RRS implementation. However, we don’t see the results of this proposal so we don’t know if the proposal is efficient or not, or what its weaknesses are. If its valid for the whole country or only for certain areas?. That makes that the proposal is not relevant in the study.

In addition, its understading is complicated. Needs clear definition and explanations, with a more approapiate theoretical framework that justifies many of the assumptions that the authors make. There are a lot of concepts throughout the text, in which the reader sometimes gets lost and does not understand where the authors want to go. Perhaps, it could be clarified with a scheme.

A real explanation of the Rural Revitalization Strategy is needed, clearer and more detailed. What is “rich life”, “livable environments” or “high degree of rural civilization defined in the RRS?”

Some other examples that need to be addressed further:

The first sentence in the Introduction said: “The problem of agriculture, rural areas and farmers […] are the inevitable products of the development of industrialization and urbanization” Why? How?  Who or what studies proves this assumption?

In page 2-3 a three methods of rural land spatial restructuring in developed countries is presented. It is generalist and vague. More bibliography is needed.

Figure 1 can lead to misunderstanding. Since 1990 there is an anual data, but the previous ones don’t, but the length on the X axis is the same. The source of the data is missing too.  

Part 2:

Why starting presenting Christaller’s theory without explaining the importance in this study or moving forwards to other theories presented later. What it is the meaning of class A, class D, etc?

Line 197 Waiter Christaller is Walter Christaller

What do the authors mean with: “sound urban development is an important condition for rural revitalization?”

What is a beautiful ecological space for the new territorial spatial planning of China?

Part 3:

Figure 2 it is very explanatory and helps to a better undesrtanding of the RRS. The figure should be explained in more detail so the reader can better understand the text.

What is the meaning of rural construction land reclamation?

What is the meaning of “high population and little land are the basic national conditions in China?” It is not understood.  

Part 4:

Line 412 CLUE-S model is the name of the software, not a type of model, it is a spatial dynamic model.

Suggestions:

First, present what is the RRS in more detail: its objectives, its strengths and weaknesses, its possible impacts.

Second, how to improve the RRS according to what other studies have said, with a strong theoretical framework (show case studies). For each objective: what other studies suggests, are these suggestions suitable or not in your study area, why? Why not?

Third, apply the methodological approach to know if it works.

Reviewer 3 Report

The paper presents a study on the spatial restructuring of urban-rural construction land based on the Rural Revitalization Strategy in China.

At the current state the manuscript seems to be a comment to the RRS and not a scientific study. It proposes solutions for calculating the village centrality index for the evaluation of villages in order to make decisions for implementation of RRS.

As a reviewer I cannot understand what is the research question and the methods to answer it. There is a theoretical framework given, but it is not clear how the study and the conclusions are related to it. It seems to be an additional explanation to the idea of spatial restructuring.

In the current form I cannot consider the manuscript to be published in “Land” Journal.

I also did some further comments as I was reading the manuscript:

General comments:

In the current state the text is often unclear due to language issues. The manuscript should be revised by an English language expert.

The reader does not necessarily know the land-use problems in China in detail (which are for sure regionally differentiated). It should be briefly and clearly introduced in the Introduction section. Neither the RRS is introduced in a clear way. The reader does not know that are the goals, measures, is there any time schedule defined, …?

There is no reflection on the goals formulated in the RRS.

Abstract:

In the current form the abstract contains only an explanation of the Rural Revitalisation Strategy (RRS), but still not naming the goals, but measures. Therefore, unclear what is really meant.

The purpose of an abstract is to give an overview of the study goals, methods, findings and conclusions.

Introduction:

RRS is mentioned a couple of times as the most important document but still it is not clear what this document says.

The introduction is missing following contents: general information on the subject, the identification of knowledge gap, new considerations (derived from the scientific literature), aim of the study/research questions, the methodological approach, brief summary of the findings.

References:

I miss references to some primary documents (e.g.: EU Documents, Christaller’s Theory, RRS, examples from Japan, Germany, …

Detailed comments:

Lines 30-31: “the three rural issues” – please explain briefly what is meant by that.  

Lines 42-43: completely different =? Than what? What do the authors mean?

Line 66:/introduction: the authors mention the RRS a couple of times, but it seems to me that this document is not directly referred to (See References)

Line 119: EU Land Remediation Guidelines – does not appear in the References

Line 236: the authors refer to the “general requirements of the RRS” but these requirements has not been introduced to the reader.

Line 420: missing reference for the given number.